# Arboviral Disease Outbreaks in the Pacific Islands Countries and Areas, 2014 to 2020: A Systematic Literature and Document Review

**DOI:** 10.3390/pathogens11010074

**Published:** 2022-01-07

**Authors:** Rosie J. Matthews, Ishani Kaluthotage, Tanya L. Russell, Tessa B. Knox, Paul F. Horwood, Adam T. Craig

**Affiliations:** 1Department of Medicine, Cairns Hospital, Cairns, QID 4870, Australia; 2Australian Institute of Tropical Health and Medicine, James Cook University, Cairns, QID 4870, Australia; ishani.kaluthotage@gmail.com (I.K.); tanya.russell@jcu.edu.au (T.L.R.); 3College of Public Health, Medical and Veterinary Sciences, James Cook University, Cairns, QID 4870, Australia; 4Vanuatu Country Liaison Office, World Health Organization, Port Vila, Vanuatu; knoxt@who.int; 5Australian Institute of Tropical Health and Medicine, James Cook University, Townsville, QID 4811, Australia; paul.horwood@jcu.edu.au; 6College of Public Health, Medical and Veterinary Sciences, James Cook University, Townsville, QID 4811, Australia; 7School of Population Health, University of New South Wales, Sydney, NSW 1466, Australia

**Keywords:** arboviral diseases, public health, surveillance, Pacific islands, dengue, Zika, chikungunya, outbreaks, low- and middle-income countries

## Abstract

Arthropod-borne diseases pose a significant public health threat, accounting for greater than 17% of infectious disease cases and 1 million deaths annually. Across Pacific Island countries and areas (PICs), outbreaks of dengue, chikungunya, and Zika are increasing in frequency and scale. Data about arbovirus outbreaks are incomplete, with reports sporadic, delayed, and often based solely on syndromic surveillance. We undertook a systematic review of published and grey literature and contacted relevant regional authorities to collect information about arboviral activity affecting PICs between October 2014 and June 2020. Our literature search identified 1176 unique peer-reviewed articles that were reduced to 25 relevant publications when screened. Our grey literature search identified 873 sources. Collectively, these data reported 104 unique outbreaks, including 72 dengue outbreaks affecting 19 (out of 22) PICs, 14 chikungunya outbreaks affecting 11 PICs, and 18 Zika outbreaks affecting 14 PICs. Our review is the most complete account of arboviral outbreaks to affect PICs since comparable work was published in 2014. It highlights the continued elevated level of arboviral activity across the Pacific and inconsistencies in how information about outbreaks is reported and recorded. It demonstrates the importance of a One-Health approach and the role that improved communication and reporting between different governments and sectors play in understanding the emergence, circulation, and transboundary risks posed by arboviral diseases.

## 1. Introduction

Arthropod-borne viral (arboviral) diseases are a significant global health problem accounting for >17% of all infectious disease cases and 1 million deaths worldwide annually [1]. Arboviral diseases are infections caused by over 100 viruses belonging to the Flaviviridae, Togaviridae, Reoviridae, Bunyaviridae, Rhabdoviridae and Orthomyxoviridae families. These viruses are spread to humans through arthropods, including mosquitos, ticks, and flies [2].

Globally, the distribution and incidence of mosquito-borne arboviruses, including Zika (ZIKV) [3,4], yellow fever [5], chikungunya (CHIKV) [6] and dengue (DENV) [7] are increasing with evidence of travel-mediated importation in areas previously unaffected, highlighting the potential for global transmission [8]. In 2020, dengue was declared the most prevalent arboviral disease worldwide, infecting approximately 219 million people and causing 400,000 deaths [9,10].

The burden of arboviral diseases is concentrated in tropical areas, where half of the world’s 7.5 billion people live [11]. In these areas, climatic conditions support populations of mosquitoes and year-round transmission of arboviruses. The combined burden of neglected tropical diseases (which include dengue, chikungunya, and Zika infections) and malaria were estimated to be 132 disability-adjusted life years lost (DALYs) per 100,000 population across the Asia-Pacific region, making these the seventh leading disease burden category [12]. A measure of burden for the Pacific Island countries and areas (PICs) was not found.

The PICs are home to 11.4 million people spread over hundreds of islands that comprise the 22 countries and areas of the region [13]. Their tropical location, topography and development status mean PICs are at increased risk of infectious disease outbreaks [13,14,15]. This is compounded by the high risk posed by natural disasters, with six PICs ranked among the top 20 countries most at risk from natural disasters (Vanuatu, Tonga, Solomon Islands, Papua New Guinea (PNG), Fiji and Kiribati) [16,17].

While arboviral outbreaks have been recorded in the Pacific since the late 1800s [18,19], reports about their occurrence are often incomplete, sporadic, delayed and lacking detail. In 2014, Roth et al. (2014) published a paper that described 28 outbreaks of dengue, chikungunya and Zika affecting PICs over the 2012–2014 period, including the co-circulation of all four serotypes of DENV [20]. Roth and colleagues concluded that the PICs were in the early stages of increasing arboviral activity that would continue for several years [20]. To the best of our knowledge, no systematic analysis of arboviral disease activity affecting the PICs has been published since 2014. We address this by providing an account of evidence of arboviral outbreak activity in the PICs between October 1, 2014 (i.e., the end of Roth et al.’s study period) and June 30, 2020.

## 2. Results

Our search of published literature identified 1176 unique peer-reviewed articles, which were reduced to 25 relevant publications when the screening criteria were applied. Our grey literature search identified 873 relevant PacNet, 48 ProMed Mail, and one WHO Disease Outbreak News posts. Collectively, the data reported 104 unique arboviral outbreaks occurring within the study period. This included 72 dengue outbreaks affecting 19 PICs, 14 chikungunya outbreaks affecting 11 PICs, and 18 Zika outbreaks affecting 14 PICs. Appendix A summarizes available information about these events and Figure 1 summarizes their temporal distribution.

### 2.1. Dengue Outbreaks

Dengue virus is a single-stranded positive-sense RNA virus of the family Flaviviridae, genus Flavivirus, primarily transmitted by *Aedes* mosquitoes. Four DENV serotypes have been identified; all can cause the full spectrum of disease [21]. WHO estimates there are 50–100 million cases and over 20,000 deaths due to dengue infection annually [22].

Dengue outbreaks of varying duration and serotype have been reported in the PICs since the late 1800s [18,19]. Historically, dengue outbreaks in PICs have had a cyclic pattern with one serotype dominating, being replaced every 3–5 years [19]; however, in 2014 Dupont-Rouzeyrol et al. reported this pattern had changed with co-circulation of serotypes becoming common [23]. Roth et al. (2014) identified 18 outbreaks of different serotypes occurring over the 2012–2014 period including, for the first time on record, all four DENV serotypes circulating at once [20,24]. Exposure to one serotype of dengue does not result in immunity to another serotype, leaving populations vulnerable to reinfection [19]. There is further evidence that those who have been infected with multiple serotypes are at increased risk of severe dengue infection [25,26].

During the study period, dengue was the most commonly reported arboviral disease; almost all (19 out of 22) PICs reported at least one dengue outbreak event. In total, our review identified 72 DENV outbreaks—15 DENV-1, 20 DENV-2, 14 DENV-3, four DENV-4, and an additional 19 outbreaks with undetermined serotype—suggesting the pattern identified by Dupont-Rouzeyrol et al. of co-circulation has continued (Figure 2).

All four dengue serotypes were co-circulating across the Pacific Island region in 2016, 2017, and 2018; and DENV-1, DENV-2, DENV-3 and DENV-unspecified were co-circulating in 2014, 2015, 2019 and 2020 (Figure 2). In the study period, six PICs recorded co-circulation of two or more DENV serotypes (Cook Islands [27], Fiji [27,28], French Polynesia [27,29], New Caledonia [27,30,31], PNG [27,28,31,32,33,34], and Solomon Islands [27,28,31,33]; and two confirmed co-circulations of all four DENV serotypes (PNG 2016 [27,28,31,32,33,34], New Caledonia 2018 [27,28]).

Dengue outbreaks ranged from less than 10 to more than 12,250 cases (due to limited testing capacity, suspected case numbers were often reported). New Caledonia was the PIC that reported the most outbreaks (*n* = 13) (Figure 2) [27,28,30,31].

Of the 72 dengue outbreaks reported, case numbers were available for 55. At least 1000 cases were reported in 18 outbreaks, and 16 outbreaks had less than 100 cases. 

During the study period, the largest dengue outbreak on record in the PICs occurred in the Solomon Islands in 2016/17. This outbreak is reported to have affected at least 12,329 people, resulting in 877 hospitalizations and 16 deaths [27,31,33]. Another notable outbreak occurred in Guam in 2019, which although small (18 cases), is considered important as it was the first autochthonous outbreak of the disease in Guam in 75 years [27].

PNG reported two separate periods of co-circulation of four and then three DENV serotypes in 2016 and 2018, respectively. The first episode of co-circulating dengue was from January to June of 2016 [27,28,31,32,33,34], followed by cases of DENV-1, DENV-2, DENV-4, and DENV-unspecified in 2018 (Figure 2) [27]. Curiously, no outbreaks were reported in PNG outside of these times. This may suggest periods of no dengue circulation; however, this seems unlikely and contradicts more recent evidence suggesting the persistent circulation of DENV-1, DENV-2, and DENV-3 in PNG for over a decade [32,33]. This is significant as endemic PNG strains serve as a potential reservoir for the spread of the virus to other countries, as demonstrated by molecular epidemiology studies that have linked outbreaks in Australia, Solomon Islands, and Fiji [32,35]. Further research is needed to understand the circulation of dengue in PNG, including elucidating the genetic diversity of strains and the potential risk for their regional dissemination. 

#### Where Information about Dengue Outbreaks Was Found

Seventy-one of the 72 dengue outbreaks were reported as situation reports and updates on PacNet, usually within days of their detection. Therefore, outbreak information was brief and limited, mostly reporting case numbers for that week for a particular disease. Twenty-five out of the seventy-two outbreaks are featured in the peer-reviewed literature. These articles were understandably published well after the events (three months to three years later), were usually published as ‘event reports’, and contained more comprehensive and richer data than PacNet reports. Less than half (*n* = 30) of the dengue outbreaks were featured on ProMed, and none were featured on WHO’s Disease Outbreak News (Figure 3).

Data across the different sources were often conflicting. For example, information on PacNet about a dengue outbreak in American Samoa in 2016 reported 809 polymerase chain reaction (PCR) confirmed cases with no mention of any other syndromic cases. However, two years later, Cotter et al. (2018) reported 3240 outbreak-associated suspected cases based on syndromic surveillance data [36].

### 2.2. Chikungunya Outbreaks

Chikungunya virus is transmitted to humans by the bite of an infected *Aedes* mosquito, most commonly *Ae. aegypti* and *Ae. albopictus*, although *Ae. polynesiensis* can also transmit the virus [40,41]. Chikungunya infections usually manifest as a self-limiting dengue-like illness with high fever, severe arthralgias, myalgias, and maculopapular rash. Rare but severe complications may occur [2,41].

Roth et al. reported seven chikungunya outbreaks between 2012 and 2014 [20]. We found 14 outbreaks in the study period affecting 11 PICs: three outbreaks in Fiji [42], two in New Caledonia [27], and one in each of America Samoa [27], Samoa [27], Tokelau [27], French Polynesia [27], Cook Islands [27,29], Kiribati [27], Marshall Islands [27], Nauru [27], and Tuvalu [27] (Figure 4).

While half of these events resulted in relatively small outbreaks (affecting <100 people), there were also notable epidemics. The most striking occurred in French Polynesia from October 2014 to March 2015, affecting an estimated 66,000 people (estimated attack rate = ~25%) [43], and resulted in 64 admissions to intensive care and 18 deaths [44]. Health authorities observed a four- to nine-fold increase in Guillain–Barre syndrome (GBS) cases at the time, raising suspicion of a causal relationship with chikungunya infection. This was the second suspected arbovirus-triggered outbreak of GBS in French Polynesia, following Zika-associated cases in 2013/14 [45]. American Samoa (2500 cases) [46], Samoa (4524 cases) [27], the Cook islands (782 cases) [27], and Tokelau (200 cases) [47] experienced chikungunya outbreaks at a similar time raising suspicion of regional spread; however, this was disproven by genomic analysis that found the French Polynesian outbreak was epidemiologically linked to the Caribbean, suggesting the infection was imported by a traveler [43].

Other notable chikungunya outbreaks were reported in Kiribati (13,309 cases) [27], the Marshall Islands (1317 cases) [27], New Caledonia, (est. 58 cases) [27], and Fiji (86 cases) [42]. Cases were also identified in Queensland (Australia), and New Zealand among travelers recently returned from a PIC [27]. 

A serological survey conducted in Fiji in 2013 by Kama et al. soon after the first chikungunya cases were detected in the country found population antibody seroprevalence rates to be low (<1%), suggesting minimal exposure had occurred and that the population was immunologically naïve to CHIKV [48]. A follow-up serosurvey conducted in June 2017 (after two outbreaks in Fiji) found a seroconversion rate of 15.5% [42].

#### Where Information about Chikungunya Outbreaks Was Found

Most data about chikungunya outbreaks were found on PacNet with 12 of 14 events (86%) reported there. Seven outbreaks were also reported in the literature [42,47,49]. Only three outbreaks were reported in ProMed and none in WHO’s Disease Outbreak News (Figure 3) [28]. 

### 2.3. Zika Outbreaks

First isolated from a non-human primate in 1947 in Africa, human Zika infections had only been reported in 14 human cases (all in Africa) until 2007 [50] when the first documented large outbreak of the disease occurred in Yap State, Federated States of Micronesia [51]. Subsequently, Zika has spread to other PICs through international travelers and trade, with Roth et al. reporting three outbreaks (in Cook Islands, New Caledonia and French Polynesia) between 2012 and 2014 [20].

Our review identified 18 Zika outbreaks affecting 14 PICs, with four countries experiencing two outbreak events during the study period (American Samoa 2016/17 and 2018, RMI 2016 and 2017, New Caledonia 2014/15 and 2016, Solomon Islands 2015 and 2016). Zika outbreaks in Tonga 2016 [27,28,33,38,39], New Caledonia 2014/15 [27,52,53], and American Samoa 2016/17 [27,28,33,39,54,55] reported relatively high case counts of 2420, 1500, and 1004, respectively (Figure 5).

Zika-associated cases of microcephaly among newborn children in French Polynesia in 2013–2014, and subsequent Zika-associated GBS in South America in late 2015/early 2016 led World Health Organization of a ‘public health emergency of international health concern’ [3,4], triggering enhanced public health and clinical response measures globally. 

In 2019, Sheel et al. linked the emergence of Zika in PICs to environmental stresses caused by climate change and associated pressures on fragile public health systems [56]. Craig et al. (2017) tested a hypothesis that in the absence of a dedicated surveillance system, routinely collected data on the incidence of acute flaccid paralysis (AFP), a cardinal sign of GBS, may provide a useful early warning indicator for Zika-related neurological complications. However, the authors found no conclusive evidence to support this hypothesis [39]. 

#### Where Information about Zika Outbreaks Was Found

A total of 14 out of 18 Zika outbreaks identified were reported on PacNet [27], 15 out of 18 outbreaks were reported in the literature [33,38,39,48,52,53,54,55,56,57,58,59], and 11 outbreaks were reported on both PacNet and in the literature. Nine outbreaks were reported on ProMed mail [28], and only one of the Zika outbreaks was reported in WHO’s Disease Outbreak News [37] (Figure 3).

Interestingly, six Zika event-related posts on PacNet reported the identification of the virus in international travelers arriving into Australia or New Zealand from a PIC where local transmission had not recently been reported [27], suggesting that ZIKV may have been circulating undetected in those countries. This highlights the value of regional disease intelligence sharing for the early detection of emergent pathogens of outbreak potential.

In 2016, Musso et al. reported the analysis of blood collected from French Polynesian blood donors. They found 42 out of 1505 asymptomatic blood donors (2.8%) were PCR positive for ZIKV [59]. This suggests that community transmission of Zika continued in French Polynesia well after communication about the outbreak ceased in mid-2014 [27].

### 2.4. Other Arboviral Disease Outbreaks 

Our review did not find definitive evidence of other arboviral outbreaks; however, the likely circulation of Ross River Virus (RRV) in French Polynesia [60] and Barmah Forest Virus in PNG [61] was mentioned in the literature. 

The suspicion of RRV circulation was supported by Aubry et al. (2017), which presents serological evidence of circulation among the French Polynesian population before 2014 [62]. Lau et al. (2017) suggest that diagnosing RRV, if present, would be unlikely as most PICs lack the capacity to perform serology to confirm infection with the virus [63]. Further, symptomatic surveillance would likely lack sensitivity given the similarity in symptom profiles with other more expected arboviral infections. With 55% to 75% of RRV infections being asymptomatic, symptom-based surveillance methods are likely inadequate [60]. 

A report by Caly et al. (2019) found evidence that Barmah Forest virus, a virus that has never been detected outside Australia, was circulating in PNG [61]. No further details about this, or other arboviral disease infections, were found in our review.

## 3. Discussion

We reviewed published and grey literature and consulted health development agencies to identify arboviral disease outbreaks occurring in PICs between 1 October 2014 and 30 June 2020 and found records of 104 unique events, including 72 dengue outbreaks affecting 19 PICs, 14 chikungunya outbreaks affecting 11 PICs, and 18 Zika outbreaks affecting 14 PICs.

We found that data on outbreaks are not routinely reported, nor are they reported in a consistent manner. The implications for this are multiple. During the acute stages of an outbreak, lack of complete and timely intelligence may delay or prevent the implementation of response measures and thereby increase the risk of disease transmission domestically. Delay in detection and response may also result in local autochthonous cases following the spread as imported cases [8]. Experience suggests that if explosive outbreaks are not responded to quickly, clinical and public health systems (particularly fragile ones) can be overwhelmed, leading to avoidable morbidity and mortality. A practical strategy to address this issue would be to establish reporting standards that stipulate the content and timeframe within which outbreak alerts ought to be made. Guidelines on what and how to share event information, currently included in the *Pacific Outbreak Manual* [64], provide a helpful resource that may be drawn upon for this. Regional public health networks, such as the Pacific Public Health Surveillance Network (PPHSN), are well placed to advocate for such an initiative and monitor compliance with agreed standards. In line with a One-Health approach, information about outbreaks that have the potential to spread between PICs ought to be shared quickly to relevant parties (government, development assistance agencies, and academics) through formal mechanisms, such as the IHR (2005), and informally through PacNet to support regional awareness. Rapid information sharing will facilitate domestic and transboundary risk assessment and, if necessary, public health action to be taken.

The value of outbreak data for evaluators and researchers would be enhanced if made publicly available through a searchable database. Such a database would serve as a longitudinal record and help track changes and trends in arboviral disease activity over time. Further, such a ‘warehouse’ for information would support an ‘After Action Review’, a practical quality improvement tool of the IHR (2005) that involves qualitative assessment of actions taken in response to an actual public health event as a means of identifying and documenting best practices, gaps and lessons. Given WHO’s role as coordinator of the PSSS, and SPC’s role as secretariat of the PPHSN, these agencies are well placed to host such a database. Further, the value of this platform could extend beyond arboviral diseases.

Our review highlights the value of PacNet, evidenced by 97 out of the 104 arboviral outbreaks (93.3%) being reported through this listserv. While, by necessity, PacNet is restricted to authorized members, the information posted is rich. The strategy suggested above (to generate a searchable database of outbreak events) could distribute critical information posted on PacNet to a global audience, providing a mechanism for enhanced intersectoral and transboundary knowledge sharing essential for national (and regional) risk assessment. Regional professional practice-aligned journals, such as *Western Pacific Surveillance and Response* (https://ojs.wpro.who.int/ojs/index.php/wpsar/index; accessed on 1 October 2021), provide another forum where outbreak reports and knowledge may be shared with a global audience.

Our analysis found discrepancies between contemporaneous case reports and later published literature that, presumably, were due to disjuncture or delays in surveillance data flows during acute outbreak events. This observation is a warning to those seeking to understand the impact of arboviral outbreak events based on incomplete case reports alone. Further, it highlights the need for an intimate understanding of local context when interpreting data, and transparency when reporting findings.

The analyzed data showed that most reported outbreaks were detected in urban settings. This is likely due to a combination of factors, including that mosquitoes responsible for the transmission of DENV, CHIKV, and ZIKV are predominantly urban-dwelling and that surveillance systems are more likely to be established in built-up areas. Extending the coverage of routine early warning surveillance to include rural areas is critical if the objectives of universal health coverage and the IHR (2005) are to be met. While important, achieving such coverage in many PICs is challenging given the infrastructure, workforce, and communication requirements. One potential solution may be to invest in event-based reporting as a strategy to complement routine indicator-based surveillance approaches. Integration of digital health ‘solutions’, including mobile phone-based data reporting and information sharing, has proven helpful in PICs [56,65] and other resource-constrained settings [66,67] and should be explored.

Surveillance is of limited value if the capacity to respond when signals are generated is lacking. Given this and the dependence on central health authorities for response in many PICs, ongoing investment in public health workforce development (particularly at local/subnational levels where outbreaks occur) is required. The PNG’s field epidemiology training program [68] and PPHSN-delivered data for decision-making courses [69] offer context-tested models of workforce development that may be applied region-wide.

Outbreak risk in PICs is exacerbated by demographic and environmental changes, including the impacts of climate change [16,17]. For example, Tropical Cyclone Winston in Fiji, 2016 [56], and Tropical Cyclone Pam in Vanuatu, 2015 [27,70], preceded outbreaks of dengue and Zika. Changes in environmental conditions due to climate change is said to contribute to change in the spatial distribution of *Aedes* mosquitos and associated risk of arboviral disease transmission [71]. Given this, arboviral disease outbreak detection should be seen as a core component of the PIC’s climate change impact mitigation efforts and not solely as an activity of health departments. Taking such a perspective may raise awareness and political will to enhance the performance of outbreak detection surveillance systems in the PICs.

This review is not without limitations. First, we relied on published data available from three sources. It is likely that many smaller outbreaks, particularly those occurring in rural and remote areas, were not detected or reported. It is also possible that some outbreaks were reported through other means, such as verbally, or via email or short-message-service, and therefore were not included in our study. Second, as confirmatory testing capacity is severely constrained in most PICs, the magnitude of outbreaks is often assessed (and reported) based on suspected case counts alone. While this likely results in some measurement bias (i.e., some false-positive reporting), we note that to rely on confirmation of pathogen in the PIC setting would likely under-represent the scale of arboviral activity and impact across the Pacific Islands. Where available, we reported both confirmed and suspected case numbers; otherwise, we make clear that the statistics presented are suspected case counts. Third, our review was limited to articles published in English and French, hence we may have missed literature published in other languages. Despite these limitations, our review provides the most complete record of published information about arboviral outbreaks to affect the PICs from October 2014 to June 2020. It will provide a useful record on which public health practitioners and researchers may draw. Our study provides the first analysis of where information about arboviral disease outbreaks in the PICs has been reported, providing insights into who, how, where, and in what timeframe outbreak intelligence is being shared.

## 4. Materials and Methods

We undertook a systematic review of published and grey literature and contacted relevant regional public health authorities to collect data about arboviral outbreaks affecting PICs. The review was undertaken in accordance with the Preferred Reporting Items for Systematic Reviews and Meta-Analyses (PRISMA) guidelines (Appendix A). We define PICs as the countries and areas located in the Pacific Ocean that fall within the WHO’s Regional Office for the Western Pacific boundary. These are American Samoa, Cook Islands, Federated States of Micronesia (FSM), Republic of Fiji, French Polynesia, Guam, Kiribati, Republic of Marshall Islands (RMI), Nauru, New Caledonia, Niue, Commonwealth of the Northern Mariana Islands (CNMI), Republic of Palau, PNG, Pitcairn Islands, Samoa, Solomon Islands, Tokelau, Tonga, Tuvalu, Republic of Vanuatu, and Wallis and Futuna. 

### 4.1. Published Literature Search

We searched electronic databases: Embase (OVID interface, 1 October 2014–30 June 2020); Global Health (OVID interface, 1 October 2014–30 June 2020); Medline (R) (OVID interface, October 1, 2014–June 30, 2020); and Medline Epub (OVID interface, 1 October 2014–30 June 2020) for published literature. The searches included three clusters of medical subject headings (MeSH) and English and French language keyword terms. The clusters related to ‘PICs’, ‘arboviral disease’, and ‘outbreaks’9 Appendix A). As it was unclear if a standard definition of an outbreak was universally applied, we relied on the author’s description of such an event. Literature was included if published between 1 October 2014 and 30 June 2020 and if it reported an arboviral outbreak occurring in at least one of the PICs within this timeframe. Literature on laboratory techniques, entomological surveillance, malaria and other parasitic diseases was excluded. The searches were performed between 2 July and 22 July 2020. RM and IK independently screened the titles and abstracts of retrieved articles; conflicts were discussed with AC and PH and a consensus was reached. We manually reviewed the reference lists of selected literature to identify additional articles. Figure 6 presents the search results.

### 4.2. Grey Literature Search

The archives of three crowd-sourced infectious disease outbreak reporting communication platforms (PacNet, ProMed mail, and the WHO Disease Outbreak News) were searched and posts related to arboviral outbreaks in the PICs between 1 October 2014 and 30 July 2020 were extracted. PacNet (www.pphsn.net/Services/PacNet/intro.htm; accessed on 22 July 2020) is the outbreak alert and communication bulletin board of the Pacific Public Health Surveillance Network (PPHSN). It is the premier digital means by which PIC infectious disease officers and regional assistance agencies share information about outbreak events affecting the islands. Of note, WHO’s weekly Pacific Syndromic Surveillance System (PSSS) bulletins are posted on PacNet; the content of these bulletins was included in our review. ProMed Mail (www.promedmail.org/; accessed on 22 July 2020) is the bulletin board of the International Society for Infectious Diseases and includes data derived from a diverse range of sources, including the media and newspapers. The WHO Disease Outbreak News (www.who.int/csr/don/en/; accessed on 22 July 2020) is an integrated global alert and response system providing information on confirmed and potential public health events of concern. Data retrieval involved systematic searches using specific PICs and arboviral diseases terms. The grey literature searches were performed between 22 July and 30 July 2020.

### 4.3. Consultation with Public Health Agency Focal Points

To triangulate data on outbreak events, we contacted representatives of three leading technical assistance agencies that support infectious disease outbreak responses in PICs: the United States Centers for Disease Prevention and Control’s Division of Vector-borne Diseases, the WHO Western Pacific Regional Office, and the Pacific Community Public Health Division.

### 4.4. Data Extraction

From each data source, the following were extracted: (i) information about the article/post/consultation (i.e., date of publication/post/consultation, full title and reference (if relevant); the type of article (i.e., research, field reports, letter/commentary, non-peer-reviewed report or correspondence, or post); and (ii) information about the outbreak event (i.e., location, timing and pathogen causing the outbreak, number of cases (by case status, if available) and deaths, method of detection and diagnosis, response measures, and any notable features of the event). Limited confirmatory testing capacity in many PICs meant reported case numbers were often based on clinical presentation or syndromic surveillance. We included information about confirmed case numbers, where reported. Data were recorded in an excel spreadsheet.

### 4.5. Data Analysis

Data were analyzed using descriptive and inductive methods to report event frequency, distribution, magnitude, and impact. Where possible, results were presented to allow comparison with those of Roth et al. (2014). Where conflict in case numbers were found, the most recently reported source of information was used. Missing data were marked as not available. 

Spatiotemporal trends of outbreaks across the study period are presented visually on a timeline and in tables, and sources of outbreak intelligence are presented in Venn-diagrams.

## Figures and Tables

**Figure 1 pathogens-11-00074-f001:**
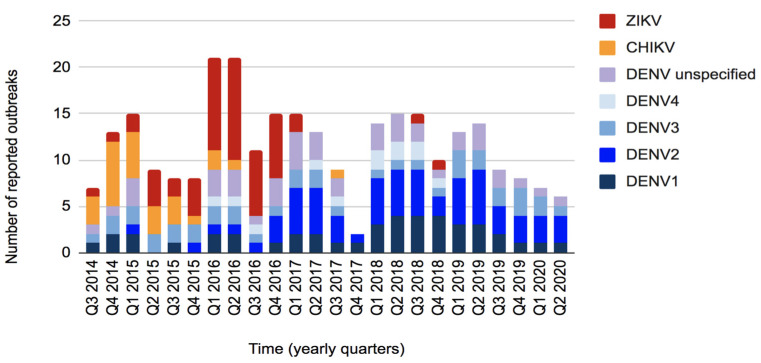
Number and temporal distribution of arboviral outbreaks, by virus, reported in the Pacific Island countries and areas, October 2014–June 2020.

**Figure 2 pathogens-11-00074-f002:**
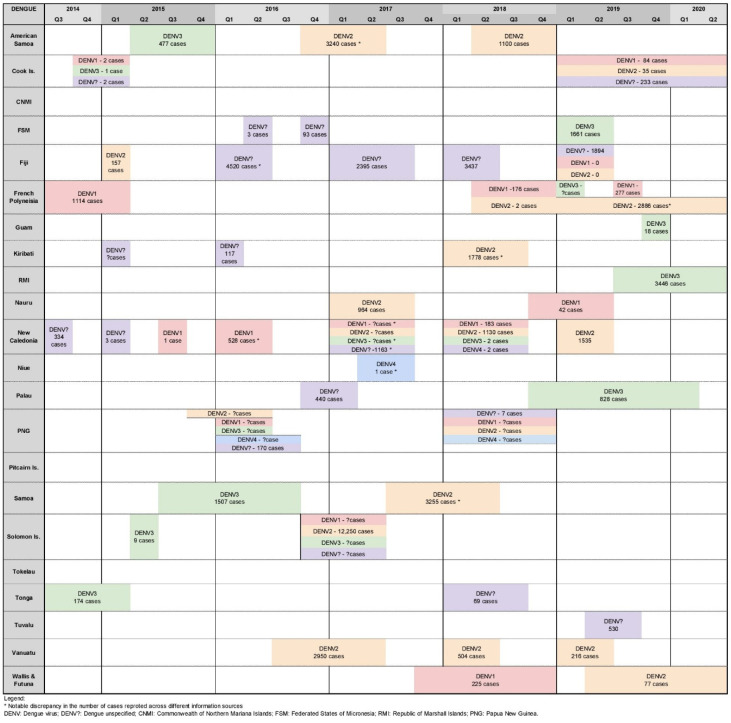
Timeline of reported dengue outbreaks (confirmed/suspected and unspecified cases) in Pacific island countries and areas, October 2014–June 2020.

**Figure 3 pathogens-11-00074-f003:**
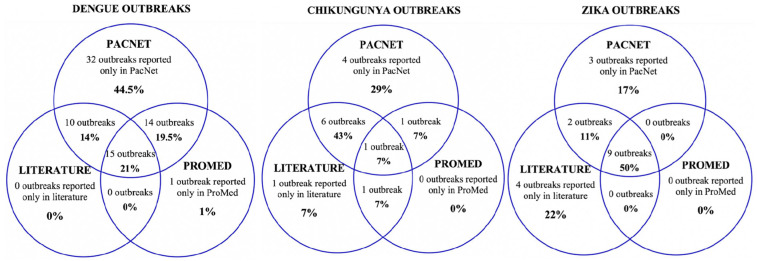
Sources of data reporting dengue, chikungunya and Zika outbreaks (confirmed/suspected and unspecified cases) in Pacific Island countries and areas, October 2014–June 2020. *Note*: In addition, one Zika outbreak was reported in WHO’s Disease Outbreak News [37]; this event was not reported on PacNet or ProMed but did feature in peer reviewed literature [38,39].

**Figure 4 pathogens-11-00074-f004:**
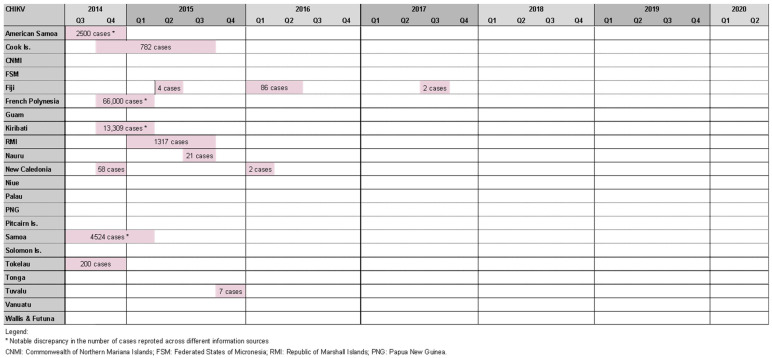
Timeline of reported chikungunya outbreaks (confirmed/suspected and unspecified cases) in the Pacific Island countries and areas, October2014–June 2020.

**Figure 5 pathogens-11-00074-f005:**
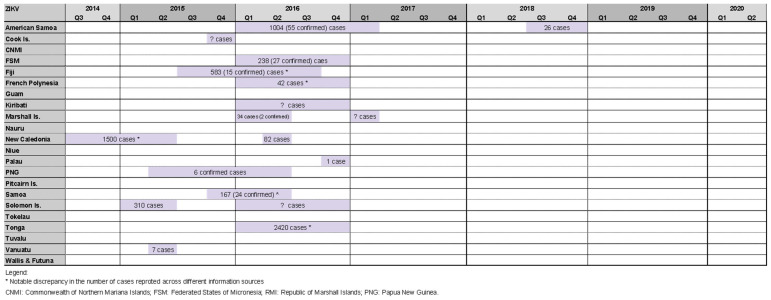
Timeline of reported Zika outbreaks (confirmed/suspected and unspecified cases) in the Pacific island countries and areas, October 2014–June 2020.

**Figure 6 pathogens-11-00074-f006:**
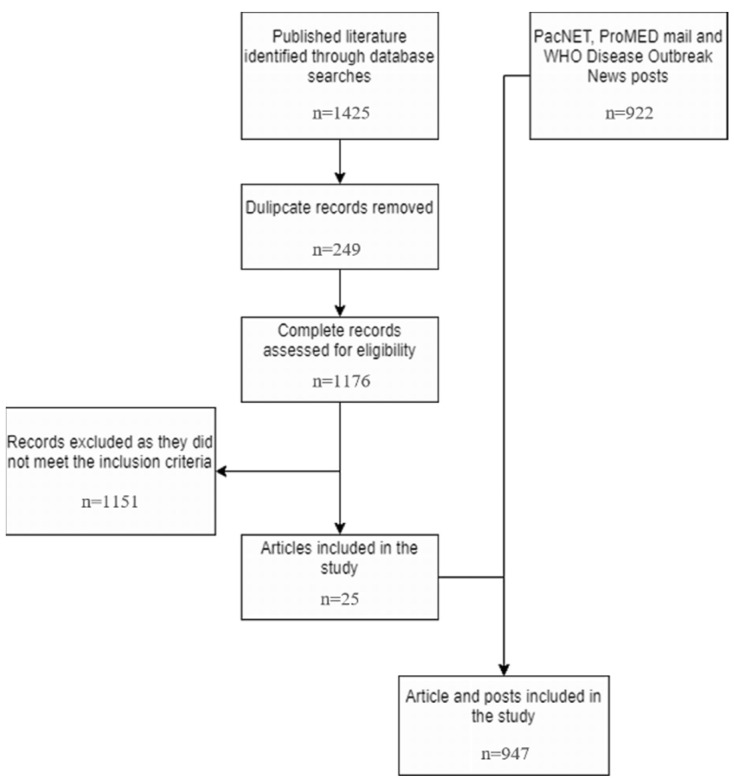
Systematic literature search diagram.

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
