# Peer review of "Arboviral Disease Outbreaks in the Pacific Islands Countries and Areas, 2014 to 2020: A Systematic Literature and Document Review"

_pathogens, 2022, doi:10.3390/pathogens11010074_

Round 1

Reviewer 1 Report

Dear Authors

This is a very interesting manuscript, which provides an update on information about arboviral outbreaks in PICs. However, I advise a careful revision to fix some typos. For example, the word “Supplementary” is misspelled throughout the manuscript, including in the “Supplementary file”. Also, you need to italicize the scientific names of the mosquitoes (Aedes should be written Aedes, etc.)

Author Response

Point 1:  This is a very interesting manuscript, which provides an update on information about arboviral outbreaks in PICs. However, I advise a careful revision to fix some typos. For example, the word “Supplementary” is misspelled throughout the manuscript, including in the “Supplementary file”. Also, you need to italicize the scientific names of the mosquitoes (Aedes should be written Aedes, etc.)

Response 1: We have reviewed the manuscript to address typos, including the misspelt word ‘supplementary.’ We have also italicised the species and genus of mosquito names.

Reviewer 2 Report

In this manuscript, the authors review relevant peer-reviewed literature and case reports to assess the frequency of Aedes-transmitted arboviruses in the Pacific Islands region between 2014-2020. These emerging pathogens are of increasing concern in this area of the world, and outbreaks in this region may precede outbreaks in other parts of the world (see the 2015-16 ZIKV outbreak in the Americas). The authors show that high levels of transmission occur throughout the Pacific, particularly of DENV, with multiple serotypes causing concurrent outbreaks in many countries. This review was clear and well written, and will be an important resource for researchers interested in the epidemiology of these viruses. I have a few minor comments:

Line 54: Suggest changing “disease category burden” to “disease burden category”

Line 87 (and throughout): Italicize “Aedes” and other Genus and species names

Line 220: Should “GBD” here be GBS?

Line 392: Change “Centre for Disease Prevention and Control” to “Centers for Disease Control and Prevention”

Fig. 3: For the ZIKV section of this figure, you may want to note that one of these cases was also reported in the WHO DON

In Fig1, the authors show a pattern of ZIKV outbreaks replacing CHIKV outbreaks from 2014-2016, before both pathogens fizzle out and are replaced by mainly DENV after 2017. I’m curious whether the authors have any thoughts on might have driven this pattern, particularly the ZIKV-CHIKV dynamics.

The authors mention some discrepancies between the contemporaneous case reports and the later published literature (lines 146-150). Can you comment on what might account for these discrepancies? Or how readers or other researchers should interpret case report data that may be incomplete after the fact?

In general, I found the text on all of the figures to be hard to read. This may just be because they were printed too small, but I would suggest making the text as large as possible to reduce this problem.

Author Response

Point 1: Line 54: Suggest changing “disease category burden” to “disease burden category”

Response 1: The script has been edited to incorporate the suggestion (at around line 60 in the revised manuscript).

Point 2: Line 87 (and throughout): Italicise “Aedes” and other Genus and species names

Response 2: Mosquito genus and species names have been italicised throughout.             

Point 3: Line 220: Should “GBD” here be GBS?

Response 3: Thank you for pointing out this typo. The script has been edited to read “GBS.”

Point 4: Line 392: Change “Centre for Disease Prevention and Control” to “Centers for Disease Control and Prevention”

Response 4: The script has been updated per the reviewer’s comment. Thank you.

Point 5: Fig. 3: For the ZIKV section of this figure, you may want to note that one of these cases was also reported in the WHO DON

Response 5: We have played around with different figure layouts and have decided that a note in the caption is the clearest way to convey that one outbreak was reported in the WHO Outbreak News.

Point 6: In Fig1, the authors show a pattern of ZIKV outbreaks replacing CHIKV outbreaks from 2014-2016, before both pathogens fizzle out and are replaced by mainly DENV after 2017. I’m curious whether the authors have any thoughts on might have driven this pattern, particularly the ZIKV-CHIKV dynamics.

Response 6: We are not aware of any evidence of interaction between these pathogens. Both CHIKV and ZIKV have a ‘boom-to-bust epidemiology’ and hence it is likely just a coincidence that ZIKV started emerging as CHIKV waned.

Point 7: The authors mention some discrepancies between the contemporaneous case reports and the later published literature (lines 146-150). Can you comment on what might account for these discrepancies? Or how readers or other researchers should interpret case report data that may be incomplete after the fact?

Response 7: In response, we have added a paragraph to the discussion (at line 312) that reads, “Our analysis found discrepancies between contemporaneous case reports and later published literature due, presumably, to disjuncture in surveillance data flows during acute outbreak events. This observation is a warning to those seeking to understand the impact of arboviral outbreak events based on incomplete case reports. Further, it highlights the need for an intimate understanding of local context when interpreting data and transparency when reporting fundings.”

Point 8: In general, I found the text on all of the figures to be hard to read. This may just be because they were printed too small, but I would suggest making the text as large as possible to reduce this problem.

Response 8: The figures have been reformatted to improve readability, including increasing the size of the text. We will take the journal’s advice if further improvement is required.

Reviewer 3 Report

First of all, I am grateful for the opportunity to review this paper. Arboviroses occur frequently over the world, the etiology is represented by a wide range of pathogens, and diagnosis must be guided also by local epidemiological conditions, including entomological studies, and laboratory data. Therefore, risk assessment and management represent a timely way for counteract those infections. In this context, the paper under review is a systematic review of published and grey literature and contacted relevant regional authorities to collect information on arboviral activity affecting Pacific Island countries during the period 2014-2020.

The article is interesting and may provide important information useful also in a one-health- perspective, but it must be improved to be suitable for an international journal, by underling the large impact of this local epidemiological data.

Title: is overstated, it can be improved.

Introduction: Local data and local epidemiological situation must not be central in the introduction. The authors should make it clear about what is the gap in the literature that is filled with this study? First of all the health risk caused by the spreading of the disease (also out of PICs) should be clearly stated, reporting also the problem of imported cases that makes of this issue an international problem (refer to Napoli C et al. Estimated imported infections of Chikungunya and Dengue in Italy, 2008 to 2011. J Travel Med. 2012; 19(5):294-7). Moreover, what is the further international contribution of the study to the literature? What are the implications of the study?

Methods: source of data and sampling procedure will benefit from more detail. PRISMA method must be used and addressed.

Discussion: I also suggest emphasizing the contribution of the study to the literature, the implications and recommendations based on previous experience and stress the concept that local autochthonous cases may spread as imported cases (refer to Napoli C et al. Estimated imported infections of Chikungunya and Dengue in Italy, 2008 to 2011. J Travel Med. 2012; 19(5):294-7). Limits section must be improved.

Author Response

Point 1: Title: is overstated, it can be improved.

Response 1: We have reviewed the title and do not feel it is overstated. We feel the title clearly and appropriately conveys the scope of the work presented and the method used. If the editor feels this is not the case, we will happily make changes and welcome any suggestions for alternative phrasing.

Point 2: Introduction: Local data and local epidemiological situation must not be central in the introduction. The authors should make it clear about what is the gap in the literature that is filled with this study? First of all the health risk caused by the spreading of the disease (also out of PICs) should be clearly stated, reporting also the problem of imported cases that makes of this issue an international problem (refer to Napoli C et al. Estimated imported infections of Chikungunya and Dengue in Italy, 2008 to 2011. J Travel Med. 2012; 19(5):294-7). Moreover, what is the further international contribution of the study to the literature? What are the implications of the study?

Response 2: We thank the reviewer for their comment. On review, we feel the introduction section of the manuscript does not unduly focus on the local epidemiology situation but instead provides context-relevant information required to understand the PIC environment and research setting.

To address the comment regarding case importation, we have edited the paragraph starting at line 52 to include relevant content and cite the suggested paper.

We note the suggestion to comment on the health risk posed by arboviral diseases. In response, we draw attention to the content at line 41 [where we write, “arthropod-borne viral (arboviral) diseases are a significant global health problem accounting for >17% of all infectious disease cases and 1 million deaths worldwide annually”] and at line 58 [where we add, “The combined burden of neglected tropical diseases (which include dengue, chikungunya and Zika infections) and malaria were estimated to be 132 disability-adjusted life years lost (DALYs) per 100,000 population across the Asia-Pacific region, making these the seventh leading disease burden category.”]

Finally, regarding the international contribution of our work, we draw attention to the last three sentences of the introduction (starting “Roth and colleagued conclude…” at line 74.) We feel this passage positions the work within the current literature, provides a rationale for the article and outlines its contribution.

Point 3: Methods: source of data and sampling procedure will benefit from more detail. PRISMA method must be used and addressed.

Response 3: A sentence has been added at around line 371 stating that the review was undertaken in accordance with the PRISMA guidelines and the 2020 PRISMA checklist added as a supplementary file.

Point 4: Discussion: I also suggest emphasising the contribution of the study to the literature, the implications and recommendations based on previous experience and stress the concept that local autochthonous cases may spread as imported cases (refer to Napoli C et al. Estimated imported infections of Chikungunya and Dengue in Italy, 2008 to 2011. J Travel Med. 2012; 19(5):294-7). Limits section must be improved.

Response 4: In response, a sentence has been added at line 283.

It is unclear what changes the reviewer suggests be made to the limitation section. We have reviewed the current text and feel it is adequate, but will be guided by the editor's/s' judgement.

Round 2

Reviewer 3 Report

The paper was improved according to my comments, and it is now suitable for pubblication